# Psychopathological Risk Assessment in Children with Hyperphenylalaninemia

**DOI:** 10.3390/children9111679

**Published:** 2022-10-31

**Authors:** Maria Cristina Risoleo, Margherita Siciliano, Luigi Vetri, Ilaria Bitetti, Anna Di Sessa, Marco Carotenuto, Francesca Annunziata, Daniela Concolino, Rosa Marotta

**Affiliations:** 1Department of Medical and Surgical Sciences, University “Magna Graecia”, 88100 Catanzaro, Italy; 2Clinic of Child and Adolescent Neuropsychiatry, Department of Mental Health, Physical and Preventive Medicine, University of Campania “Luigi Vanvitelli”, 81100 Caserta, Italy; 3Associazione Anatolia, 81040 Curti, Italy; 4Oasi Research Institute-IRCCS, Via Conte Ruggero 73, 94018 Troina, Italy; 5Department of Woman, Child and of General and Specialized Surgery, Università degli Studi della Campania “Luigi Vanvitelli”, Via Luigi De Crecchio 2, 80138 Naples, Italy

**Keywords:** phenylketonuria, neuropsychiatric disorders, psychopathological risk, CABI, SAFA

## Abstract

Background: Phenylketonuria (PKU) is a rare congenital disorder caused by decreased metabolism of phenylalanine determining cerebral impairments. If untreated, PKU might lead to intellectual disability, seizures and behavioral disorders. The aim of this study is to provide a characterization of the psychopathological profile of a pediatric population diagnosed with PKU at newborn screening. Methods: an accurate neuropsychological evaluation of 23 patients (aged 8–18 years) with hyperphenylalaninemia (defined as experimental group, EG) and in 23 age-matched healthy controls (defined as control group, CG) was performed using the Child and Adolescent Behavior Inventory (CABI) and Self-Administrated Psychiatric Scales for Children and Adolescents (SAFA) questionnaires. Results: the CABI test showed significant differences for the sub-scales related to “Irritable mood”, “Oppositional-provocative symptoms” and “ADHD” in the EG compared to CG (*p* = 0.014, *p* = 0.032, and *p* = 0.032, respectively). Patients with hyperphenylalaninemia also presented with significant differences both for anxiety disorder scale and depression scale of SAFA test than controls (*p* = 0.018 and *p* = 0.009, respectively). Conclusions: children and adolescents with early diagnosis of PKU showed a psychopathological risk profile characterized by an increased risk of experiencing symptoms such as mood deflection, anxiety, attention deficit, oppositional defiant behavior, and obsessive traits than healthy peers. Our findings highlighted the need of the inclusion of a neuropsychiatric evaluation in the management of these patients to improve their overall quality of life.

## 1. Introduction

Phenylketonuria (PKU) is a rare congenital autosomal recessive disorder of metabolism of phenylalanine (Phe) affecting intellectual function [1,2], firstly described in 1934 in two siblings with “phenylpyruvic oligofrenia” [3]. PKU is caused by phenylalanine hydroxylase deficiency due to the presence of pathogenic variants in the *PAH* gene. To date, approximately 1000 PAH mutations have been identified as causative agents of PKU [1,4,5,6].

The phenylalanine hydroxylase (PAH) catalyzes the hydroxylation of phenylalanine to tyrosine in a reaction requiring the co-substrate tetrahydrobiopterin (BH4). Enzyme deficiency resulted in phe accumulation with subsequent hyperphenylalaninemia, increased phenylketones (namely phenylketonuria), and in reduced myelin formation and dopamine, norepinephrine, and serotonin production [7,8,9,10]. As a consequence, these changes mainly affected brain functionality [10,11].

HPA is the primary biochemical abnormality of PKU, in which normal blood phe concentrations (35–120 μmol/L) are exceeded. Less severe forms of PAH deficiency are variously referred to as mild HPA, or benign HPA (phe concentrations 120–600 μmol/L), mild PKU (phe concentrations pretreatment 600–1200 μmol/L), moderate PKU (phe concentrations 900–1200 μmol/L), while severe forms are referred to as classic PKU (phe concentrations >1200 μmol/L). Historically, different severities of PAH deficiency have been differentiated by untreated phe concentration [1], providing data on the overall prevalence of this different severity PAH deficiency. However, these prevalence data must be carefully interpreted, since establishing the severity of PAH deficiency is difficult because patients are now diagnosed before the biologically highest untreated phe concentration is reached [12].

Estimates reported a similar incidence both in Europe (1:10,000 live births) and United States (1: 15,000 live births) [5].

Given the severe and irreversible intellectual disability closely related to untreated PKU, most countries included newborn screening to avoid the dramatic neurological burden (including developmental delay, seizures, afinalistic movements, athetosis, electroencephalographic abnormalities, behavioral and psychiatric disorders) of the disease [1].

Early treatment enables patients to have normal intelligence, although some neurological impairments such as slightly lower IQ values than the general population [13,14,15] and alterations in executive functions have been reported in PKU patients [16,17].

Notably, several studies have reported impairments in a series of specific domains such as attention [18], processing speed [19], working memory [20], motor skills and coordination [21].

Moreover, patients with PKU might experience a large spectrum of neuropsychiatric disorders (ranging from psychomotor agitation to autistic traits) [22,23,24] with different phenotypes by age and various degrees of severity based on phe blood levels [22,25].

The psychological stress associated with the chronic disease and its management (including dietary restrictions and long-term follow-up) plays a pathogenic role for social and emotional disorders that could arise in these patients [26]. A decrease in the quality of life and psychological wellbeing seems to be a constant following the various phases of patients with PKU life: childhood, adolescence, adulthood [26,27]. We aimed to assess the psychopathological risk of a cohort of school-aged children diagnosed with PKU at newborn screening.

## 2. Materials and Methods

The total sample consisted of 46 Italian children and adolescents, of these, 23 subjects belong to the experimental group consecutively attending the Genetics Department of the Pediatric Clinic of the “Magna Græcia” University between 1 January 2020–30 June 2020. Informed consent was obtained from all subjects involved in the study. The study was conducted in accordance with the Declaration of Helsinki. Ethical approval was obtained from the Ethics Committee of Calabria Region Central Area Section (no. 2021/107).

### 2.1. Study Population

Twenty-three patients with hyperphenylalaninemia were included in the experimental group (EG), while 23 healthy subjects were considered as control group (CG). The main features of the study population are shown in Table 1. Patients included in the EG met the following criteria: age 8–18 years, early PKU diagnosis through neonatal screening program; normal cognitive functioning. Exclusion criteria were considered as follows: age <8 or >18 years, intellectual disability, anorexia nervosa, and presence of other chronic diseases. Among patients with hyperphenylalaninemia, seventeen followed a low-protein diet with the support of a specialist dietician. Seven out of twenty-three patients also received a drug therapy (sapropterin). Moreover, post-screening tests did not confirm the PKU diagnosis in nine patients of this group but revealed a mild hyperphenylalaninemia.

To be eligible for the healthy control group, the following criteria were considered: age 8–18 years, absence of PKU diagnosis, of intellectual disability, of anorexia nervosa, and of other chronic diseases. The CG was recruited in elementary, middle, and high school.

Out of the 23 enrolled patients in the EG, nine patients showed mild hyperphenylalaninemia, and among these, six did not require particular dietary restrictions, while three required the introduction of protein-free foods into the diet as the presence of non-optimal phe values.

In the remaining fourteen patients, post-screening tests confirmed the diagnosis of PKU. They had a balanced low-protein diet and were supported by a specialist dietician. Seven patients also received a medical treatment with sapropterin dihydrochloride.

The European guidelines for the optimization of phenylketonuria (PKU) treatment highlight that diet is the cornerstone of treatment, although some patients may benefit from tetrahydrobiopterin (BH4). No action is required if the blood phenylalanine concentration is less than 360 μmol/L (6 mg/dL). For higher values, the treatment must last a lifetime. Target treatment concentrations are as follows: less than 360 μmol/L (6 mg/dL) for individuals aged 0–12 years and less than 600 μmol/L (10 mg/dL) for individuals older than 12 years. The minimum requirements for the management and follow-up of patients with PKU are planned based on age, treatment adherence, and clinical status. Nutritional, clinical, and biochemical follow-up are necessary for all patients, regardless of therapy. In this study, diet adherence and metabolic control were good [28].

The phe values in the EG ranged between 2 and 13 mg/dL, with an average of 5.5 mg/dL. This blood value was the last measurement.

### 2.2. Evaluation Tools

#### 2.2.1. CABI

The CABI (Child and Adolescent Behavior Inventory) questionnaire is a feasible and rapid test for parents evaluating various emotional and behavioral problems that may arise in developmental age. As psychopathological risk test assessment, it represents a useful tool for clinicians to explore possible emotional and behavioral problems [29].

It has been released in CPEMH in 2013 [30]. In one study [31], the predictive validity of the CABI questionnaire for clinical diagnosis, compared to the scales oriented to the Diagnostic and Statistical Manual of Mental Disorders (DSM) of the Child Behavior Checklist (CBCL) was evaluated. The accuracy values obtained (probability of correct classification) were high for both instruments and significantly better for the CABI anxiety and attention deficit hyperactivity disorder (ADHD) scales and for the CBCL scales for oppositional defiant disorder (ODD) and conduct disorder (CD); no significant differences were found for the depression scales. All areas under the curve (AUC) of the analysis of the operational characteristics of the receiver reached excellent values, suggesting an excellent predictive capacity of the five scales of the two instruments. Comparison of the AUC showed that the anxiety and ADHD scales of CABI give significantly higher values than those of CBCL, indicating that these two scales have better predictive capacity [31]. The questionnaire consisted of 75 items exploring the following psychopathological domains (sub-scales): questions 1–4 explore somatic symptoms; questions 5 to 10 explore anxiety; question 11 explores phobias; questions 12–15 explore obsessive–compulsive symptoms; questions 16–17 explore insecurity; question 18 explores post-traumatic stress disorder; questions 19–28 explore depressive symptoms; questions 29–32 explore irritable mood; questions 33–37 explore oppositional-defiant symptoms; questions 38–42 explore conduct disorders; questions 43–51 explore ADHD; questions 52–55 explore reality assessment; questions 56–61 explore social relations; question 62 explores enuresis; question 63 explores encopresis; question 64 explores bulimia; questions 65–67 explore anorexia nervosa; questions 68–69 explore sex; questions 70–72 explore substance abuse; questions 73–74 explore school; and question 75 explores bullying. Specific cut-off scores of the 10 areas with the most relevant items (questions) have been pointed out. Regarding the score, 2 points have been assigned to the answers “True”, 1 point to “Partly or sometimes true”, and 0 point to “False”. For the other 11 areas, if the answer was “True” or “Partly and sometimes true”, parents were asked for further data [29].

#### 2.2.2. SAFA

All the enrolled subjects were also tested for the self-administration scales for children and adolescents (SAFA). The SAFA battery is a tool that can be used for individuals aged between 8 and 18. Its purpose is to give the neuropsychiatrist a first investigation into the psyche of the subject to guide the subsequent clinical examination depending on the results.

The questionnaire consisted of six scales such as the “A” scale for anxiety, the “D” scale for depression, the “O” scale for obsessive–compulsive disorders, the “P” scale for psychogenic eating disorders, the “S” for somatic symptoms, the “F” scale for phobias. Furthermore, to adapt the questionnaire to the child’s comprehension skills for the “A” scale, based on the school age, three distinct questionnaires were provided (8–10 years; 11–13 years; 14–18 years), and two questionnaires for the scales “D”, “O”, “S” and “P” in (8–10 years; 11–18 years). As a descriptive scale, the “F” scale for phobias detects and assesses 24 different types of fear. The items of each scale have been formulated to be sufficiently indicative for the symptom investigated and to be easily understood. For each item, three possible answers are allowed: “true” (replaced by “often” in the somatic symptom scale), “false”, and “a middle ground”. Within each scale, it is then possible to identify sub-scales examining the different clinical phenotypes of the disorder to provide an initial definition of the psychic profile of the subject. It was also possible to identify items defined as “critical” as the responses provided to them by the subjects belonging to the healthy population varied greatly with respect to the corresponding responses provided by the subjects suffering from mental disorders. The time required to complete the questionnaire generally varies between 30 and 60 min. Scores must be attributed according to the instructions relating to the individual scales. For scales A and D (including positive and negative items), the appropriate grids are necessary to calculate the total score and of the individual subscales. For the other scales, the calculation of the total score is simpler and does not require a grid, since all items are negative [32].

### 2.3. Statistics

Quantitative data were expressed as mean ± standard deviation (SD), while categorical data were presented as frequency.

The distribution of the different variables was examined, and the appropriate parametric or nonparametric test was used. Differences in continuous variables were determined by using the independent sample t-test for normally distributed continuous variables and the Kruskal–Wallis test in case of non-normality. Data not-normally distributed were log-transformed before the analysis, but raw means are shown. The association of both CABI and SAFA domains with EG and CG was also tested after adjustments for age and sex.

*p*-values < 0.05 were considered statistically significant. Statistical analyses were performed using STATISTICA 8.0 software (StatSoft).

## 3. Results

The mean age of the EG and of the CG were 13.21 ± 3.46 years and 12.69 ± 3.44 years, respectively. According to the CABI test, patients belonging to the EG presented with values above the cutoff in the following domains: Somatic symptoms sub-scale (seven patients); “Anxiety” sub-scale (five); “Depressive symptoms” sub-scale (seven); “Irritable mood” sub-scale (nine); “Oppositional-Defiant Symptoms” sub-scale (seven); “Conduct Disorders” sub-scale (two); “ADHD” sub-scale (four); “Evaluation of Reality” sub-scale (one); “Social Relations” sub-scale (three); “Anorexia Nervosa” sub-scale (two).

A comparison between the number of subjects showing results above the cut-off in the CABI sub-scales between the EG and CG is shown in Figure 1.

Subjects included in the CG showed values above the cutoff in the following CABI domains: Somatic Symptoms sub-scale (three subjects); “Anxiety” sub-scale (two); “Depressive Symptoms” sub-scale (four); “Oppositional-defiant symptoms” sub-scale (three); “Conduct Disorders” sub-scale (two); “Social relations” sub-scale (one). Regarding the “Mood Irritable”, “ADHD”, “Reality Assessment” and “Anorexia Nervosa” sub-scales, all the enrolled subjects in this group had values within the normal range.

CABI domains between the two groups were compared (Table 2). Significant differences were found in the EG for the sub-scales “Irritable mood”, “Oppositional-defiant symptoms” and “ADHD” compared to the CG (*p* = 0.014, *p* = 0.032, and *p* = 0.032) (Table 2). Both ADHD and Anorexia nervosa domains showed a significant association also after adjustments for age and sex (*p* = 0.010 and *p* = 0.05, respectively) (Table 3).

The administration of the SAFA test in the EG resulted in a pathological score for the following domain: “Somatic symptoms and hypochondria” (Scale S) (one patient); “Obsessive Compulsive Symptoms” scale (Scale O) (one). Moreover, borderline scores were recorded in this group as follows: “Scale A” (Anxiety) (nine patients); ”Scale S” (Somatic symptoms and hypochondria) (four); “Scale D” (Depression) (ten); “O Scale” (Obsessive Compulsive Symptoms) (five).

SAFA test in the CG found no pathological scores. With respect to the borderline scores, the following scales were affected in this group: “Scale A” (three subjects); “Scale S” (one), “Scale D” (two), and “Scale O” (one).

A comparison between the number of subjects showing borderline or pathological scores in the SAFA scales between the EG and CG is shown in Figure 2.

By comparing the results of the SAFA test between the two groups, significant differences emerged in the “Scale A” for anxiety disorders and in the “Scale D” for depression (*p* = 0.018 and *p* = 0.009, respectively) (Table 4). Similarly, the EG showed higher scores for “depressed mood”, “irritable mood”, and “sense of guilt” than control group (*p* = 0.011, *p* = 0.021, and *p* = 0.019, respectively). Scores for the “order” sub-scales of the “O Scale”, and for the “acceptance of one’s body” and “anorexic behaviors + acceptance of the own body” of the “P scale” were significantly higher in the EG compared to the CG (*p* = 0.017, *p* = 0.009, *p* = 0.0041, respectively) (Table 4). Significant associations of SABA domains with EG and CG after adjustments for age and sex are shown in Table 5.

## 4. Discussion

Our study provided evidence for a neuropsychiatric profile in a population of school-aged children with PKU diagnosis at newborn screening. In fact, our results confirmed the role of PKU as risk factor for both behavioral and emotional disorders in developmental age.

Over the past decades, studies have demonstrated various neuropsychiatric manifestations in patients with PKU [2]. As a direct PKU consequence, pediatric patients might experience neurological impairments closely related to the interference with neurodevelopmental processes and acute or chronic dysfunctions of excitatory/inhibitory or monoaminergic neurotransmitters [1,33].

Several studies focused on the presence of externalizing problems in patients with PKU, but the introduction of both dietary recommendations and early restriction of phe intake also brought internalizing problems [34,35].

As previously described [36,37,38], our findings reported a higher proportion of anxiety and depression symptoms in children with PKU. In fact, a large amount of data showed that children suffering from PKU presented with specific features such as greater fears, feelings of being different, low self-esteem, sadness, withdrawal, lack of autonomy, depressed mood, anxiety, physical disorders or social isolation [39,40,41,42]. Notably, depressed mood and anxiety have also been found as the most commonly encountered symptoms in adulthood [22,43,44,45].

Therefore, a general model characterized by the presence of internalizing symptoms has been largely recognized in these patients [34,35].

In addition to the neurological damage directly caused by the PKU, the psychological stress associated with its management (including dietary restrictions and long-term follow-up) might also play a pathogenic role for social and emotional disorders encountered in these patients [34]. Of note, studies comparing PKU and diabetes mellitus as chronic diseases reported higher rates of internalizing problems such as depressed mood, anxiety, physical disturbances or social isolation [37].

Furthermore, in a 2014 research study, patients with PKU were compared to those with congenital hypothyroidism, by demonstrating higher rates of internalizing symptoms related to parental inadequate emotional adaptation to underline diseases and maladaptive coping strategies in both groups [46]. Taken together, these findings supported a role of follow-up and rigorous dietary treatments for psychological distress not only in PKU patients but also in their families [46]. Conversely, the larger prevalence of internalizing disorders in children and families strictly adherent to dietary recommendations could be explained by the greater awareness of potentially developing symptoms of anxiety and depression [47].

The neuropsychiatric burden highlighted in our cohort also encompassed obsessive traits in patients with hyperphenylalaninemia, as demonstrated by the higher values in the subscales of “order” and “poor acceptance of the body”. This could be explained by the prevalent anxiety reported in PKU patients and the need for strict compliance to dietary restrictions. Moreover, the poor acceptance of the body observed in our population might be also due to the critical age (ranging from pre-adolescence to adolescence) of the enrolled subjects facing further serious concerns about their chronic condition [48].

According to previous findings [18,49,50], further behavioral impairments such as oppositional-defiant symptoms and ADHD have been found in our population. In particular, ADHD symptoms have been closely related to the phe exposure timing [51,52]. A potential pathophysiological link between PKU and ADHD might be represented by lower dopamine levels, especially in the prefrontal cortex and in the striatum of these patients [51]. Some limitations deserve to be mentioned. Firstly, our sample size is small. The lack of a follow-up does not allow us to evaluate long-term outcomes of these patients. However, the presence of a control group might enhance the strength of our results. Moreover, both dietary restriction and adherence to metabolic control were only indirectly examined through blood phenylalanine values. However, the phe concentrations of our population seemed to suggest a good adherence, but caution is needed in interpreting these findings as the availability of one single value per patient. Another limitation is that the experimental group is made up of individuals treated differently, ranging from strict dietary treatment to drug treatment and moderate dietary restriction to no treatment, and no internal comparison was carried out between the subjects of the clinic group treated with different modalities. However, realistically, the subjects of the experimental groups shared the same psychological stress associated with chronic disease and its management (including long-term follow-up and adaptations of restrictive dietary regimes or the possible pharmacological treatment in case of inadequate metabolic control).

## 5. Conclusions

In addition to the well-known neurological impairments, children and adolescents with an early PKU diagnosis presented with a psychopathological risk profile. Notably, PKU patients have a greater predisposition to developing anxiety, depression, attention deficit, oppositional-defiant symptoms and to manifest obsessive traits compared to the general population.

In this perspective, a neuropsychological evaluation should be included in the management of PKU to further improve the psychophysical well-being of these patients, as also suggested for other rare conditions [53].

## Figures and Tables

**Figure 1 children-09-01679-f001:**
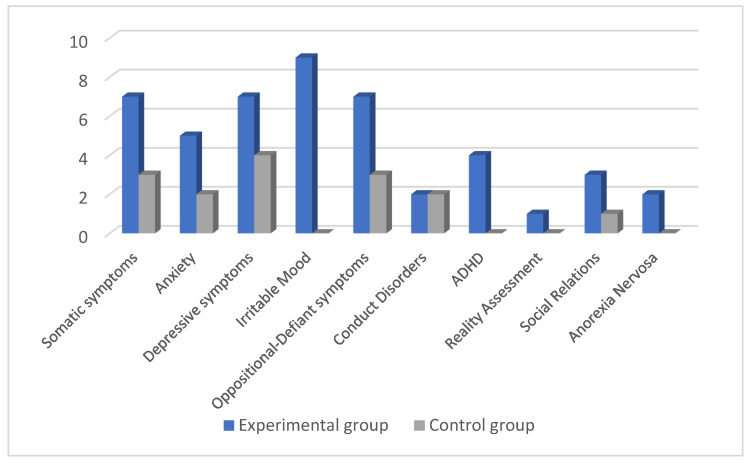
Number of subjects showing results above the cut-off in the CABI sub-scales in the EG and CG.

**Figure 2 children-09-01679-f002:**
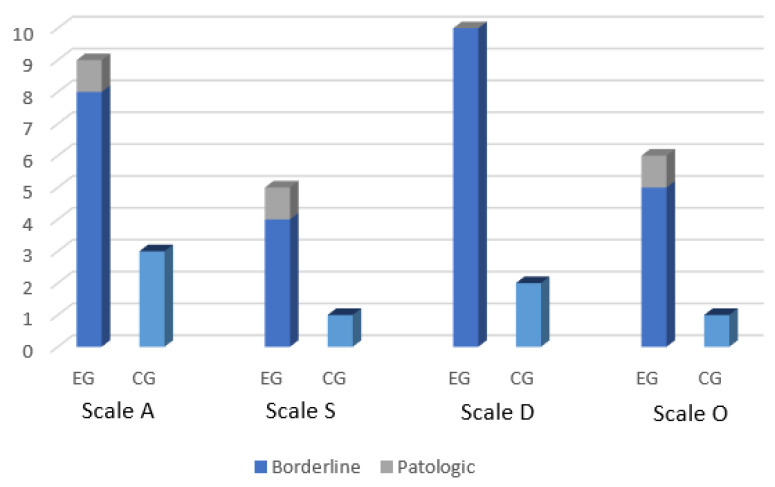
Number of subjects showing borderline or pathological scores in the SAFA scales in the EG and CG.

**Table 1 children-09-01679-t001:** Main features of the study population.

	EG (*n* = 23)	CG (*n* = 23)
Male/Female (No.)	7/16	5/18
Mean Age (years, SD)	13.21 (3.46)	12.69 (3.44)
Mild Hyperphe (No., mean ± SD blood Phe)	9 (3.88 ± 1.05)	-
PKU (No., mean ± SD blood Phe)	14 (6.5 ± 3)	-
Total	23 (5.5 ± 2.78)	

SD: standard deviation

**Table 2 children-09-01679-t002:** Comparison of CABI domains between EG and CG.

CABI Test	Experimental Group		Control Group		*p*-Value
	Mean	SD	Mean	SD	
Somatic symptoms	1.35	1.61	1.17	1.07	0.882
Anxiety	2.70	2.49	3.04	1.89	0.376
Depressive symptoms	2.87	3.70	1.39	1.62	0.206
Irritable mood	2.09	2.48	0.43	0.73	**0.014**
Oppositional-defiant symptoms	1.83	2.48	0.61	1.41	**0.032**
Conduct disorders	0.22	0.52	0.09	0.29	0.367
ADHD	2.78	3.37	0.83	1.47	**0.032**
Reality assessment	0.13	0.46	0.00	0.00	0.153
Social relations	0.61	1.78	0.13	0.34	0.866
Anorexia nervosa	0.35	0.65	0.09	0.29	0.110

Bold values denote statistical significance at the *p* < 0.05 level.

**Table 3 children-09-01679-t003:** Age-and sex-adjusted analyses of CABI domains between EG and CG.

Domain	*p*-Value	R^2^	Adjusted R^2^
ADHD	0.010	0.23	0.17
Anorexia nervosa	0.05	0.16	0.10

**Table 4 children-09-01679-t004:** Comparison of the SAFA domains between EG and CG.

SAFA Test		EG		CG	*p*-Value
	Mean	SD	Mean	SD	
**Scale A**	47.96	8.93	41.35	7.76	**0.018**
Generalized anxiety	50.91	10.95	45.43	8.89	0.087
Social anxiety	49.74	10.75	45.35	7.57	0.141
Separation anxiety	45.35	11.24	39.17	5.12	0.071
School-related anxiety	45.70	9.05	42.65	8.18	0.137
**Scale S**	43.74	10.08	38.22	5.45	0.086
Somatic symptoms	43.78	10.22	38.35	5.02	0.121
Hypochondria	43.04	13.01	41.91	7.18	0.330
**Scale D**	50.30	11.36	41.70	5.95	**0.009**
Depressed mood	49.78	9.05	43.43	6.07	**0.011**
Anhedonia	48.87	9.13	44.96	4.77	0.141
Touchy mood	53.74	10.17	46.26	8.93	**0.021**
Sense of inadequacy and low self-esteem	49.48	13.04	42.09	5.51	0.078
Insecurity	49.17	12.13	44.43	9.17	0.144
Guilt	49.13	11.99	41.17	8.28	0.019
Hopelessness	49.57	15.54	44.48	4.29	0.083
**Scale O**	43.96	16.59	39.04	7.52	0.546
Obsessive thoughts	44.04	15.52	39.26	5.85	0.550
Compulsions	46.13	10.30	41.52	6.04	0.183
Rupophobia	43.91	11.46	39.48	8.41	0.146
Order and control	49.09	15.63	41.30	8.35	**0.017**
Doubt and indecision	48.26	13.68	42.17	6.97	0.222
**Scale P** Bulimic behavior	44.35 46.26	8.67 7.20	42.16 43.30	7.55 4.09	0.629 0.181
Anorexic behavior	44.22	6.93	44.04	8.67	0.446
Acceptance of one’s own body	43.00	6.83	38.70	3.84	**0.009**
Anorexic behavior + Acceptance of one’s own body	44.00	8.42	39.70	6.81	**0.041**
Fear of maturity	44.57	8.87	44.91	9.66	0.899
Perfectionism	41.83	13.41	43.22	14.93	0.921
Inadequacy	46.57	9.04	41.30	4.86	0.056

Bold values denote statistical significance at the *p* < 0.05 level.

**Table 5 children-09-01679-t005:** Age-and sex-adjusted analyses of SAFA domains between EG and CG.

Domain	*p*-Value	R^2^	Adjusted R^2^
**Scale A**	0.05	0.16	0.10
Separation Anxiety	0.05	0.16	0.10
School related anxiety	0.002	0.29	0.24
**Scale S**	0.006	0.25	0.20
Somatic symptoms	0.016	0.21	0.16
Hypochondria	0.05	0.16	0.10
**Scale D**	0.016	0.21	0.15
Depressed mood	0.035	0.18	0.12
Touchy mood	0.05	0.16	0.10
Sense of inadequacy and low self-esteem	0.018	0.21	0.15
Guilt	0.018	0.21	0.15
**Scale O**	0.014	0.22	0.16
Rupophobia	0.04	0.17	0.11
Order and control	0.009	0.24	0.18
Doubt and indecision	0.003	0.27	0.22
**Scale P**	0.028	0.22	0.16
Acceptance of one’s own body	0.039	0.17	0.12

## Data Availability

The data that support the findings of this study are available from the corresponding author upon reasonable request.

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
