# Peer review of "Psychopathological Risk Assessment in Children with Hyperphenylalaninemia"

_children, 2022, doi:10.3390/children9111679_

Round 1

Reviewer 1 Report

With regards the methodology:

1. Is it worth stating that the CABI tool is a validated and established tool used since xxx dates - I am thinking that it is validated and established?

2. SAFA tools - I am less clear that all sections of this are validated - the authors say that the F scale for Phobias is unique (does that mean they  just designed it only for this study? ).  The authors then omit all mention of the F scale findings.  So the reader is left asking why is this?

3. The control group appears to be younger than the experimental group by 4 years - so for the SAFA questionnaires on anxiety, I am thinking that the Control group had very few study subjects completing the 14-18 group questionnaire - could this have affected results, having a younger control group?  Or were the results "pooled" for the EG and CG before comparison and so the age specific questionnaires not so relevant?

Author Response

We thank the Editor and the Reviewers for the time spent in reviewing our manuscript. All the changes have been highlighted in red in the text.

1. Is it worth stating that the CABI tool is a validated and established tool used since xxx dates - I am thinking that it is validated and established?

Answer: Thank you for your comment. The Child and Adolescent Behavior Inventory (CABI) is a questionnaire designed to collect information from the parents of children and adolescents for the screening, epidemiological studies, and clinical evaluation. It has been released in CPEMH in 2013. (Cianchetti C., Pasculli M, Pittau A, Campus MG, Carta V, Littarru R et al. Child and Adolescent Behavior Inventory (CABI): Standardization for Age 6-17 Years and First Clinical Application.Clin Pract Epidemiol Ment Health. 2017) and its predictive validity for clinical diagnosis compared to the scales oriented to the Diagnostic and Statistical Manual of Mental Disorders (DSM) of the Child Behavior Checklist (CBCL) has been tested. In particular, the accuracy values ​​obtained (probability of correct classification) were high for both instruments and significantly better for the CABI anxiety and attention deficit hyperactivity disorder (ADHD) scales and for the CBCL scales for oppositional defiant disorder (ODD)) and conduct disorder (CD); no significant differences were found for the depression scales. All areas under the curve (AUC) of the analysis of the operational characteristics of the receiver reached excellent values, suggesting an excellent predictive capacity of the five scales of the two instruments. Comparison of the AUC showed that the anxiety and ADHD scales of CABI give significantly higher values ​​than those of CBCL, indicating that these two scales have better predictive capacity. (Cianchetti C.,Faedda N., Pasculli M., Ledda M.G., Diaz G., Peschechera A. et al. Predictive validity for the clinical diagnosis of a new parent questionnaire, the CABI, compared with CBCL. Clin Child Psychol Psychiatry. 2020 Apr;25(2):507-519). Please see lines 144-155 of the revised version of the text.

  1. SAFA tools - I am less clear that all sections of this are validated - the authors say that the F scale for Phobias is unique (does that mean they just designed it only for this study?).  The authors then omit all mention of the F scale findings.  So the reader is left asking why is this?

Answer: thank you for your valuable comment. Unfortunately, a statistical evaluation for SAFA-F scale has not possible to perform as it produces qualitative values exploring patient’s phobias. Please see lines 180-181 of the revised text.

  1. The control group appears to be younger than the experimental group by 4 years - so for the SAFA questionnaires on anxiety, I am thinking that the Control group had very few study subjects completing the 14-18 group questionnaire - could this have affected results, having a younger control group?  Or were the results "pooled" for the EG and CG before comparison and so the age specific questionnaires not so relevant?

Answer: thank you for your observation. Indeed, there was a typo in the text. Both groups were aged 8-18 yr. The mean age for the experimental group (EG) was 13.21 ±3.46 SD, while for the control group (CG) was 12.69± 3.44SD. Please see lines 113, 209-210 of the revised text.

Reviewer 2 Report

The authors present an evaluation of behaviour and psychiatric abnormalities in patients with early diagnosed PKU compared to healthy controls. They conclude that PKU patients are at an increased risk for development of behavioural and emotional problems. Whereas it is evident that early identification and treatment prevents physical and cognitive impairment in patients with PKU the impact of treatment on psychological well-being is still unclear. Therefore this publication could have contributed to the ongoing discussion. Unfortunately the study design suffers from major structural problems.

1.  The study population of 23 individuals („experimental group“) is a mixed cohort of patients with mild hyperphenylalaninemia (MHP), BH4-responsive PKU and mild or classical PKU. Most of the patients are on a protein restricted diet, but 6 patients do not follow any dietary restrictions. Unfortunately, the authors do not give any information about the Phe-tolerance of individual patients. Anthropometric data such as BMI are missing, yet „anorexia nervosa“ is one of the items evaluated in the CABI test. It is evident that dietary requirements are highly variable depending on the PKU subtype and BH4 responsiveness. Therefore the different subgroups have to be evaluated seperately, which is not done. However, as the cohort is already quite small statistical evaluation would then have been difficult.

2. The authors discuss that neuropsychiatric problems may also be caused „as a direct PKU consequence“. With that they most likely mean poor metabolic control. Dietary adherence and metabolic control are never even mentioned in the whole manuscript and it is not possible to evaluate them in the setting of this study. Phe concentrations in the study population „ranged between 2 and 13 mg/dl“. The authors do not describe if this number is the last value measured or if it is the mean of Phe concentrations throughout life or during the last year. As the age range of the study group is between 8 and 18 years and in most European countries target ranges differ in age groups from 8 to 12 and from 13 to 18 years dietary adherence and metabolic control may be good, moderate or insufficient, but no rating is performed by the authors.

3. The control group is not sufficiently defined. The only requirement for this group is the absence of PKU and intellectual disability. It is not mentioned if individuals in this cohort have other chronic diseases (which may be possible in a hospital-recruited cohort). It is also not clear why the age range differs from the „experimental group“ (8-14 versus 8-18 years).

4. The authors should have cited more recent references, such as DOI: 10.1016/j.ymgmr.2018.10.002 or doi: 10.1002/jmd2.12202. Most references they use date many years back.

5. The introduction is quite poor and does not provide sufficient background. PKU ist not „usually“ but always caused by PAH deficiency. Defects of tetrahydrobiopterin metabolism are completely different disorders but are still mentioned, whereas the different PKU subtypes are neither described nor defined. Line 48/49: The sentence about the higher PKU incidence in Turkey and Northern Ireland is irrelevant to this study.

6. As neither the extent of dietary restriction nor adherence of metabolic control was surveyed the discussion is a collection of speculations instead of an evaluation of study results.

Author Response

We thank the Editor and the Reviewers for the time spent in reviewing our manuscript. All the changes have been highlighted in red in the text.

The authors present an evaluation of behaviour and psychiatric abnormalities in patients with early diagnosed PKU compared to healthy controls. They conclude that PKU patients are at an increased risk for development of behavioural and emotional problems. Whereas it is evident that early identification and treatment prevents physical and cognitive impairment in patients with PKU the impact of treatment on psychological well-being is still unclear. Therefore, this publication could have contributed to the ongoing discussion. Unfortunately, the study design suffers from major structural problems.

  1. The study population of 23 individuals („experimental group“) is a mixed cohort of patients with mild hyperphenylalaninemia (MHP), BH4-responsive PKU and mild or classical PKU. Most of the patients are on a protein restricted diet, but 6 patients do not follow any dietary restrictions. Unfortunately, the authors do not give any information about the Phe-tolerance of individual patients.

Anthropometric data such as BMI are missing, yet “anorexia nervosa“ is one of the items evaluated in the CABI test. It is evident that dietary requirements are highly variable depending on the PKU subtype and BH4 responsiveness. Therefore, the different subgroups have to be evaluated separately, which is not done. However, as the cohort is already quite small statistical evaluation would then have been difficult.

Answer: thank you for your comment. Our study showed an increased risk of developing behavioral and emotional problems in PKU patients than general population regardless of diet emphasizing the urge need for early identification, monitoring, and treatment of PKU to prevent both physical and cognitive impairments disease-related. We specified dietary requirements of our cohort. Please see lines 113-134 of the revised text.

Regarding anthropometric data, mean BMI of the experimental group was of 21.56±4.60 SD, while in the control group this information was not available but no patients with anorexia nervosa were enrolled, as specified in the revised text. Please see lines 107,113,114 of the revised version of the text.

  1. The authors discuss that neuropsychiatric problems may also be caused „as a direct PKU consequence“. With that they most likely mean poor metabolic control. Dietary adherence and metabolic control are never even mentioned in the whole manuscript and it is not possible to evaluate them in the setting of this study. Phe concentrations in the study population „ranged between 2 and 13 mg/dl“. The authors do not describe if this number is the last value measured or if it is the mean of Phe concentrations throughout life or during the last year. As the age range of the study group is between 8 and 18 years and in most European countries target ranges differ in age groups from 8 to 12 and from 13 to 18 years dietary adherence and metabolic control may be good, moderate or insufficient, but no rating is performed by the authors.

Answer: following your valuable comment, we specified these aspects in the revised version of the text.  Phe concentrations in the study population "ranged between 2 and 13 mg / dl". This value represents the last measurement. Please see lines 123-134 of the revised text.

  1. The control group is not sufficiently defined. The only requirement for this group is the absence of PKU and intellectual disability. It is not mentioned if individuals in this cohort have other chronic diseases (which may be possible in a hospital-recruited cohort). It is also not clear why the age range differs from the „experimental group“(8-14 versus 8-18 years).

Answer: The control group included healthy subjects without any chronic diseases recruited in a school setting. The age range was not different between the two groups (8-18 years), as previously incorrectly stated (there was a typo). Please see lines 107,108, 113-115, 209,210 of the revised text.

  1. The authors should have cited more recent references, such as DOI: 10.1016/j.ymgmr.2018.10.002 or doi: 10.1002/jmd2.12202. Most references they use date many years back.

Answer: thank you for your suggestion. We added the aforementioned references in the revised version of the manuscript. Please see the introduction and the discussion sections of revised version of the text.

  1. The introduction is quite poor and does not provide sufficient background. PKU ist not „usually” but always caused by PAH deficiency. Defects of tetrahydrobiopterin metabolism are completely different disorders but are still mentioned, whereas the different PKU subtypes are neither described nor defined. Line 48/49: The sentence about the higher PKU incidence in Turkey and Northern Ireland is irrelevant to this study.

Answer: following your valuable comment, we revised and improved the introduction section accordingly. Please see lines 4-7, and 50-67 of the revised text.

  1. As neither the extent of dietary restriction nor adherence of metabolic control was surveyed the discussion is a collection of speculations instead of an evaluation of study results.

Answer: following your valuable comment, we revised the discussion accordingly. Adherence to metabolic control was investigated by providing blood phenylalanine values in the study (please see line 131 of the revised text). In the revised version of the text, we also underlined as a limitation that dietary restriction and adherence to metabolic control were investigated only indirectly through blood phenylalanine values. Please see lines 314-316 of the revised text.

Round 2

Reviewer 2 Report

The manuscript is improved but still needs extensive editing.

General: 

1. The major problem of the study is the fact that the experimental group consists of individuals with different treatment modalities ranging from strict dietary treatment to drug treatment and moderate dietary restriction to no treatment at all. Therefore, the psycholocical burden is very different between these groups, but the cohort is only discussed as a whole without any differentiation. This point still needs revision. 

2. Throughout the study the authors refer to the experimental group as "Patients with PKU". However, 9 patients formally do not fall into the category "PKU" but have "mild hyperphenylemia". Therefore the wording should be "Patients with hyperphenylalaninemia" instead. 

Introduction:

3. Line 38: The sentence "PKU is also known as phenylalanine hydroxylase deficiency..." needs rephrasing: "PKU is caused by phenylalanine hydroxylase deficiency due to the presence of pathogenic variants in PAH gene".

4. As stated in the previous review defects of tetrahydrobiopterin metabolism are completely different disorders. As the study is about patients with PAH deficiency only, detailed description of these disorders (line 50-57) is not necessary and confuses the reader. This paragraph, as well as line 71 should be omitted. 

Materials and Methods:

5. Line 108: "The PKU group followed a low protein diet" is misleading. Does this mean that only the PKU patients followed a diet and the MHP patient did not? Or does this descibe the experimental group in general? Then it would be incorrect as 6 patients did not require dietary restrictions (line 116). 

5. Line 116: Only 23 "patients" were enrolled, not 46. Individuals in the control group should not be described as "patients". 

6. Line 121/122: The sentence "Seven patients were also treated with the Kuvan drug" should be rephrased. 

Discussion

7. The authors state that the patient's Phe concentrations suggest "good adherence" but present only one single value per patient. This limitation should be discussed.

8. Indeed the need to adhere to dietary restrictions could explain the neuropsychiatric manifestations described. However, six patients did not follow any dietary recommendations. Did these patients perform better in the tests? This point should be taken up and discussed.

Author Response

Dear Reviewer,

I would like to thank you for your valued comments and suggestions to the article. As you requested, we made all the necessary changes in our manuscript to address your concerns and we detailed below how the points raised have been accommodated. The main changes are highlighted in yellow in the text of the manuscript. We kept the changes made during the first revision written in red in the text of the manuscript. From the changes made in the revised manuscript and responses provided below, I hope you are convinced that we have adequately addressed the reviewer’s concerns and made the paper better. If there are any further questions, please feel free to let me know.

The manuscript is improved but still needs extensive editing.

General: 

  1. The major problem of the study is the fact that the experimental group consists of individuals with different treatment modalities ranging from strict dietary treatment to drug treatment and moderate dietary restriction to no treatment at all. Therefore, the psychological burden is very different between these groups, but the cohort is only discussed as a whole without any differentiation. This point still needs revision. 

Answer: Thanks for the suggestion. The authors believe that, although the experimental group is made up of individuals treated with different modalities ranging from strict dietary treatment, to drug treatment and moderate dietary restriction to no treatment, they share psychological stress, associated to chronic disease and its management (including long-term follow-up and adaptations of restrictive dietary regimes or the possible pharmacological treatment in case of inadequate metabolic control). However, as you suggested the psychological burden is probably different between these subcategories, therefore we added this consideration in the limitation of the study (321-331)

  1. Throughout the study the authors refer to the experimental group as "Patients with PKU". However, 9 patients formally do not fall into the category "PKU" but have "mild hyperphenylemia". Therefore the wording should be "Patients with hyperphenylalaninemia" instead. 

Answer: following your comment, we modified the wording "Patients with PKU” as “Patients with hyperphenylalaninemia" throughout the revised version of the text. We modified the title paper accordingly too.

Introduction:

  1. Line 38: The sentence "PKU is also known as phenylalanine hydroxylase deficiency..." needs rephrasing: "PKU is caused by phenylalanine hydroxylase deficiency due to the presence of pathogenic variants in PAH gene".

Answer: we rephrased this sentence according to your comment. Please see lines 38-39 of the revised text.

  1. As stated in the previous review defects of tetrahydrobiopterin metabolism are completely different disorders. As the study is about patients with PAH deficiency only, detailed description of these disorders (line 50-57) is not necessary and confuses the reader. This paragraph, as well as line 71 should be omitted. 

Answer: following your comment, we deleted lines suggested and the  aforementioned paragraph in the revised version of the text.

Materials and Methods:

  1. Line 108: "The PKU group followed a low protein diet" is misleading. Does this mean that only the PKU patients followed a diet and the MHP patient did not? Or does this describe the experimental group in general? Then it would be incorrect as 6 patients did not require dietary restrictions (line 116). 

Answer: we agree with your comment. We clarified it in the revised text, please see lines 113-114

  1. Line 116: Only 23 "patients" were enrolled, not 46. Individuals in the control group should not be described as "patients". 

Answer: we used “subjects” instead of “patients” for individuals belonging to the control group.

  1. Line 121/122: The sentence "Seven patients were also treated with the Kuvan drug" should be rephrased.

Answer: we rephrased the sentence accordingly.

Discussion

  1. The authors state that the patient's Phe concentrations suggest "good adherence" but present only one single value per patient. This limitation should be discussed.

Answer: we agree with your observation and we are aware that only one single value per patient does not allow to establish the adherence in an accurate manner. We added this as a limitation. Please see lines 321-331 of the revised text.

  1. Indeed the need to adhere to dietary restrictions could explain the neuropsychiatric manifestations described. However, six patients did not follow any dietary recommendations. Did these patients perform better in the tests? This point should be taken up and discussed.

Answer: Thanks for the indications, this is a limitation. Unfortunately, in this study, an internal comparison was not carried between the subjects of the clinic group treated with diet or without dietary recommendations. We added your consideration in the limitations of the study (lines 321-331)